# Phytotherapeutic Approaches in Canine Pediatrics

**DOI:** 10.3390/vetsci11030133

**Published:** 2024-03-20

**Authors:** Fausto Quintavalla

**Affiliations:** Dipartimento di Scienze Medico Veterinarie, Università di Parma, 43126 Parma, Italy; fausto.quintavalla@unipr.it

**Keywords:** puppy, phytotherapy, adverse reaction, herbal remedies, dog, holistic medicine, herbal medicine, phytocomplex

## Abstract

**Simple Summary:**

When dealing with fragile subjects such as newborns, veterinarians can face clinical challenges. Data on the safety and efficacy of many medications used in puppies are surprisingly scarce. As a result, puppies are sometimes given ineffective medications or medications with unknown harmful side effects. This has prompted many professionals, and even more dog owners, to turn to the use of herbal remedies because of their easy access, convenience, greater tolerability, and wide margin of safety, thus leading them to becoming more popular worldwide. However, natural = safe is not always the case. The purpose of this review is to illustrate, based on the limited bibliography available, the areas of application of phytotherapeutic drugs in canine pediatrics while also pointing out the limitations of this therapeutic practice.

**Abstract:**

Phytotherapy is a clinical modality that incorporates botanical remedies as part of the therapeutic approach. It is a very ancient branch of medicine that is currently undergoing a renaissance, evident in the numerous preparations available on the market. The majority of these formulations are for preventive and curative use in adult animals. Experimental experiences in the pediatric age group are particularly scarce within the existing literature. Since these products are readily accessible, dog owners often turn to them due to their ease of availability, a preference for self-medication, and the perception that herbs are safer, gentler, and less expensive than conventional medications, often leading them to bypass seeking the advice of experienced professionals. The purpose of this review is to illustrate, on the basis of the currently available bibliography, phytotherapeutic approaches in canine pediatrics, paying particular attention to the adverse effects resulting from the use of certain plants, even when used in conjunction with some synthetic drugs. Consequently, it becomes evident that further clinical and more relevant studies, specifically focusing on puppies, are needed to increase knowledge about the effects of herbal remedies.

## 1. Introduction

Non-conventional medicine, also known as holistic medicine, encompasses several specialties within veterinary medical practice, including homeopathy, homotoxicology, oligotherapy, flower therapy, phyto-aromatherapy, as well as the more recognized traditional Chinese medicine, acupuncture, chiropractic therapy, and manual therapies [1]. Phytotherapy, in particular, is a discipline that scientifically defines the use of medicinal/officinal plants in clinical practice, with deep historical roots, especially in farm animal health [2,3]. So-called ethnoveterinary medicine, referring to animal health, is based on traditional folk medicine, which, primarily utilizing plants, offers a cost-effective alternative to medications, particularly in rural areas [4]. In the Mediterranean basin, approximately ten major botanical families are utilized in veterinary folk phytotherapy [5,6], but their number is much higher in other regions worldwide [7,8].

The inclusion of nutraceuticals in the therapeutic approach and/or diet of companion animals has gained popularity in recent decades, resulting in a wide variety of supplements and foods on the market containing a wide type of plant extracts, pure compounds, and, more generally, functional compounds [9]. This trend may probably be attributed to the fact that many pet owners personally use such remedies, appreciating their benefits. Moreover, they prefer the oral route due to its lower invasiveness, physiological compatibility, ease of administration, and better compliance from dogs. In doing so, however, they often bypass the guidance of veterinarians, thus failing to truly evaluate the efficacy of such natural remedies and, more critically, increasing the risk of adverse reactions.

Many modern pharmaceuticals are natural products or are derived from plants, and advances in technology and manufacturing could facilitate the production of plant-based vaccines and therapies. The plant constitutes a well-defined therapeutic entity, with all its parts being utilized, including the leaves, whole roots, stem bark, root bark, seeds, flowers, bulbs, fruits, latex, borneol, and malt [10,11]. It is worth noting that over 60% of allopathic drugs are derived from plants and herbs (Table 1), and the major difference between a synthetic drug and a herbal medicine lies in the distinct mode of action between a single isolated active ingredient and a comprehensive phytocomplex.

The phytocomplex is a complex and polymorphic biochemical entity. It represents the pharmacological unit of the medicinal plant acting through the complementary and synergistic activity of its individual constituents [12]. However, it is worth remembering that, for example, the *Cannabis* plant, depending on its genetics and various synthase activities, can produce over 100 different cannabinoids [13].

The active ingredients variably present in the phytocomplex include alkaloids, glucosides, lipids, essential oils, antibiotics, resins, carbohydrates, mineral salts, tannins, and vitamins. Thus, a vegetable drug is essentially a concert of active ingredients, and its effects result from the integrated action of the multiplicity of its constituent substances. Clinical and pharmacological studies have shown that it is not possible to trace the action of a phytotherapeutic remedy solely to an isolated and purified active ingredient separated from the regulatory substances. As a result, their combined application remains largely intuitive. It is worth noting that analytical technology is available to provide accurate identification and quantification of each constituent in a product. Nevertheless, recent studies have analyzed a random assortment of oils and found poor correlation between the composition indicated on the label and the actual content in some products, across both human and veterinary markets [13].

Different formulations are prepared with various plant components, mainly for oral and topical administration, in the form of decoction, powder, crushed, paste, concoction, poultice, juice, infusion, etc. [14,15]. In recent years, pharmaceutical scientists have shifted their attention to designing drug delivery systems for herbal medicines using a scientific approach that allows for the development of new formulations, such as nanoparticles, microemulsions, matrix systems, solid dispersions, liposomes, solid lipid nanoparticles, and so on [16]. Natural plant-derived compounds have been intensively studied as potential antiglycating, neuroprotective, and antioxidant agents, and they have been exploited for their antimicrobial, antifungal, antiparasitic, anti-inflammatory, antioxidant, and soothing properties [17,18]. It is important to remember that oxidative stress associated with inflammation leads to significant changes in the structure and function of biomolecules [19,20], and it is no coincidence that cannabis, aloe, thyme, artemisia, and milk thistle are the most commonly prescribed natural products [21].

In clinical practice, particularly in dogs, phytotherapy is used for various purposes, especially, as regards Spain, by young female veterinarians working in clinics [21]. From careful bibliographic research, it is highlighted that the phytotherapeutic treatments were carried out in the course of external otitis [22], respiratory tract infections [7], treatment of heart disease [23], lower urinary tract diseases [24], hepatopathy [25], prevention of plaque formation, gingivitis, and periodontal disease [26], prevention of motion sickness and chronic vomiting diseases [7], treatment of perianal adenopathy, rheumatoid arthritis, joint pain, and joint cartilage injuries [7,27,28], treatment of helminthiasis [7,29], tick infestations and demodectic mange [30,31], management of canine appendicular osteosarcoma to increase patient survival time [32], certain oculopathies [33,34], management of fireworks-related fear [35], post- and peripartum situations [8,36], and various other pathological conditions. Some works report on the use of medicinal herbs for hunting purposes [37,38]. The vast majority of these studies have been performed on adult animals, while experiences in pediatric age are very limited. 

## 2. Use of Herbal Medicine in Puppies

The term pediatric generally refers to the first 6 months of life, although dogs are often referred to as puppies up to one year of age. A further subdivision provides for the distinction into neonates, aged from 0 to 2 weeks; infants, aged from 2 to 6 weeks; and juveniles, aged 6 to 12 weeks [39]. In dogs, the neonatal period up to weaning presents significant challenges for pediatric veterinarians and breeders due to the various potential threats during this stage (Table 2). This period is associated with high morbidity and mortality rates, which can range from 5.7% to 35% [40]. The incidence of mortality tends to decrease gradually over time. According to Lawler [41], the mortality rate decreases from 49.6% during days 0–3 (live births) to 16.9% within days 4–28, further dropping to 1.2% within days 29–42, and reaching 2.4% within days 42–45 (post-weaning).

The therapeutic management of puppies is conditioned by their anatomical and functional peculiarities, which are somewhat analogous to those in human medicine. In these patients, certain barriers are less effective. For instance, the blood–brain barrier is not fully developed, allowing substances to pass into encephalic structures. The skin barrier, characterized by thin skin, less subcutaneous fat, and higher water content, enhances the capacity to absorb xenobiotics. Despite some similarities, there are differences between the two species. In fact, there is often a tendency to equate a 2-year-old child with a 6-week-old puppy, questioning the real validity of experimentation on young animals in drug development [44].

In both children and puppies, pharmacokinetic and pharmacodynamic parameters differ from those in adults, contributing to both the lack of therapeutic effect and the different drug response [45,46,47]. Metabolic differences between species could also play a more important role in defining the disparities between drug bioavailability in humans and animals [48]. This is particularly relevant considering that the key steps in the fate of a synthetic drug, whether parenterally or orally administered, from absorption, distribution, metabolism, and excretion, are qualitatively and quantitatively different in the young and adult (Table 3 and Table 4).

The different degrees of development in target organs, the ontogeny of metabolizing enzymes, and transporters vary in young animals. In general, drug metabolism is deficient in neonatal animals, and thus drugs can hardly be changed into inactive and excretable forms. Next to metabolic degradation, the renal excretory capacity is decisive in the termination of drug action. In newborn subjects, biotransformation processes mediated by the microsomal system of drug-metabolizing enzymes (involving redox reactions and conjugation with glucuronic acid) are rather insufficient. In addition to differences in body composition, the greater volume of distribution for most drugs in newborns is related to a greater fraction of free drugs available for distribution due to a lower binding to plasma proteins (Figure 1).

Oral drug absorption, however, is also associated with the nature of the dosage form administered (e.g., tablet, capsule, solution), and oral bioavailability is influenced by drug size, lipophilicity, charge, and binding to dietary constituents [55]. For these reasons, if possible, drug therapy should be avoided in patients under 30 days of age. After this point in time, the dose for adults may be given per unit of body weight. When quantitative data are not available, the initial dose may be increased in regard to the greater volume of distribution. In general, since drug elimination is delayed, the dose interval should be lengthened (or the maintenance dose decreased) to avoid cumulative effects [56]. All of these modalities lead to an individualized and more tolerable therapeutic approach in pediatric patients; phytotherapy seems to partially meet these conditions.

Some of the important bioactive phytonutrients include polyphenols, terpenoids, resveratrol, flavonoids, isoflavonoids, carotenoids, limonoids, glucosinolates, phytoestrogens, phytosterols, anthocyanins, ω-3 fatty acids, and probiotics, which would find wide application in the prevention and/or treatment of diseases and various physiological disorders in pediatric patients. However, in pediatrics, complementary and alternative medicine (CAM) may exert multiple physiological activities, respectively modifying the safety profile of CAM and the severity of related adverse reactions [57]. Confirming the skepticism about the use of CAM, including that of herbal medicine, in pediatric patients, it appears that in Italy only 10% of childrens aged 0–14 were treated with CAM, and 2% with phytotherapies [6]. Unfortunately, data on the percentage of phytotherapy use in puppies compared to adult animals are lacking.

Based on the available literature, the use of phytotherapy in canine pediatrics is mainly limited to treating infectious and parasitic gastroenteritis, infectious respiratory diseases, and dermatological conditions. Furthermore, it is worth noting that most pediatric studies in this field are conducted on a small number of subjects.

### 2.1. Gastrointestinal Disorders

The main experimental experiences concern the use of phytotherapeutics in both bacterial and viral gastroenteritis.

Herbal medicines, as a result of their well-known antioxidant properties, find ample justification in young dogs with parvovirosis, where the role of oxidative stress is evident [58] and their beneficial effects on the immune response to bacterial attack in weaning puppies is obvious [59], as also recently confirmed by Chetan et al. [60] in evaluating the role of nutraceuticals combined with pharmacological therapy in puppies aged between 1.5 and 6 months affected by parvovirus. Essential oils of eucalyptus, lemongrass, and thyme have demonstrated antiparvovirus canine type 2 (CPV-2) activity in vitro [61]. The antioxidant action of several phytocomplexes (e.g., *Curcuma longa*, *Aloe vera*, *Boswellia serrata*, *Triticum aestivum*, *Plantago* spp., *Serpylli herba*, *Vaccinium myrtillus*, *Camellia sinensis*, and *Citrus*) find indication in the treatment of acute and chronic enteropathies [62]. Xaxa and Kumar [63] have reported the antioxidant efficacy of *Centella asiatica* in puppies with gastroenteritis at a dose of 10 mg/kg/day PO for 7 days. Centella contains tannins, flavonoids, triterpene saponins, essential oil, phytosterols, polyphenols, and amino acids that are exploited for their therapeutic action on blood circulation, and for this reason it is also used for external use in the treatment of ulcers and fissures.

*Curcuma longa* is a perennial herb that, when dried, becomes the source of the spice turmeric and the flavonoid curcumin, which show strong antioxidant properties, inhibit the development of autophagosomes in colonic epithelial cells, and also have an anti-inflammatory effect in acute and chronic inflammatory processes [64].

A methanolic extract of *Lannea coromandelica* bark, at doses of 100 and 200 mg/kg body weight, is also able to significantly reduce the frequency and severity of diarrhea. The antidiarrheal property of *Lannea coromandelica* is mediated by inhibiting hypersecretion, gastrointestinal motility, and increasing gastric transit time [65].

A commercial phytotherapeutic solution, consisting of thyme, Icelandic lichen, hyssop, and soapwort root, used in combination with licorice extract (50 mg/day), administered concurrently with robenacoxib, is able to reduce the risk of NSAID-induced colonic mucosal hyperemia in young Beagle dogs [66].

In clinical situations characterized by vomiting and diarrhea, slippery elm has been proposed for its emollient and soothing properties on the digestive tract. Marshmallow root creates a mucin-like coating on irritated tissues, as well as having antibacterial activity against pathogenic bacteria and immunostimulatory effects. Pill treatment (*Kang ning wan*) has also been shown to have an effect against protozoan infections [1].

Functional constipation is a common pediatric problem in puppies and children. Some botanical species such as senna, the fruit or leaf of the plant *Senna alexandrina* or *Cassia augustifolia*, and the dried bark of the trunk or branches of the plant *Cascara sagrada* (*Rhamnus purshiana*) have been used in the latter but may cause side effects such as abdominal cramps and discomfort.

One study reported that *Aloe vera* juice, of which leaves are full of milky brown or yellowish juice containing the most bioactive compounds, was used as a gastric tonic for vomiting in dogs [29].

According to the literature review, there is a great diversity in species of which extracts have shown potential anthelmintic activity. New anthelmintics are needed for the pharmacological arsenal because the drugs currently available are not entirely satisfactory [67]. Some nutraceuticals, for example, made from garlic, eucalyptus, and *Ruta graveolens*, as well as white mustard (*Sinapis alba*), green walnut leaves, and nasturtium seeds, are known for their vermifuge properties. Celery, coriander (*Coriandrum sativum*), ginger (*Zingiber officinale*), and cayenne pepper are recommended for worm prevention in pets. Arecholine and several other alkaloids obtained from the dried seeds of *Areca catecu* have been used to treat tapeworm infestations in dogs [7]. Aqueous extracts of areca nut and pumpkin seeds in the experimental treatment of puppies affected by heterophyasis have provided promising results, but further investigations are needed [68].

### 2.2. Respiratory Diseases

In puppies, canine infectious respiratory disease complex (CIRDC) involving multiple viral and bacterial pathogens can be complicated by bronchopneumonia, resulting in more severe signs such as dyspnea, weight loss, pyrexia, and even death [69]. Some phytotherapeutics can contribute to clinical improvement and recovery.

Infusions of sage and licorice roots with the addition of honey seem to be effective for cough control. Essential oils obtained from bramble leaves, flowers of elderberry (*Sambucus nigra*), borage (*Borrago officinalis*), and thyme (*Thymus vulgaris*) also have a soothing effect on cough. Various herbal supplements (*Gan mao ling*, *Ehr chen wan*, frittilary, and medlar syrup) have also been proposed to reduce coughs associated with pharynx and upper trachea irritation as well as suppress expectoration [1].

During bronchitis, pleurisy, and pneumonia, garlic and eucalyptus have been indicated for their antibacterial activity [7]. *Echinacea purpurea* powder (1:3) administered with food (1.0 g/10 kg/day for 8 weeks), due to its ability to induce nonspecific stimulation of the immune system, showed significant improvement at week 8 in 92% of dogs with manifestations of chronic and seasonal upper respiratory tract infections, including pharyngitis/tonsillitis, bronchitis, and kennel cough [70].

### 2.3. Dermatopathies

Skin diseases are one of the most common reasons owners take their dogs to the veterinarian.

Four medicinal plants, marigold (*Calendula officinalis* L.), St. John’s Wort (*Hypericum perforatum* L. agg.), chamomile (*Matricaria chamomilla* L., syn. *Matricaria recutita* L.,) and sage (*Salvia officinalis* L.), are those most commonly used in pyoderma, canine atopic dermatitis, external otitis, wounds, and dermatophytosis in dogs due to their broad-spectrum antibacterial and antifungal actions, as well as their anti-inflammatory and other beneficial effects on healthy skin [71].

In traditional medicine, certain plants are often used in the management of different cutaneous ectoparasitosis, e.g., celery (*Anethum graveolens*), cumin (*Carum curvi*), coriander (*Coriandrum sativum*), bay leaf (*Laurus nobilis*), peppermint (*Mentha piperita*), virgin tree (*Sassafras albidum*), chamomile (*Matricaria chamomila*), quassia wood (*Quassia amara*), and parsley. A spray of juniper (*Juniperus communis*) essential oil and water appears to be an effective flea repellent for use in dogs and cats, although this plant, if ingested, can be dangerous to pets [72]. Neem oil, in the form of shampoos and sprays, often in combination with other herbs (e.g., citronella, eucalyptus), has an insect repellent effect, as well as having an anti-inflammatory and antimicrobial action [1]. The essential oil of *Tagetes minuta*, which includes more than 27 components, including limonene (13.0%), piperitenone (12.2%), and α-terpinolene (11.0%), is also reported to have acaricidal action in puppies. Indeed, a 20% solution of *Tagetes minuta* (Asteraceae) oil in 2% Tween was found to be 100% effective against larvae, nymphs, and adults of *Rhipicephalus sanguineus* (Acari: Ixodidae) [30].

A lotion prepared with garlic, lemon, elderberry leaves (*Sambucus niga*), violet leaves (*Viola odorata*), wormwood (*Artemisia absinthium*), and clover (*Trifolium pratense*) would be effective against mange [7]. A liquid solution made from the rhizomes of *Acorus calamus*, *Cedrus deodara*, *Eucalyptus globulus*, *Azadirachta indica*, and *Pongamia pinnata* has been shown to be effective and well-tolerated in juvenile forms of demodicosis, and it is considered a viable alternative to common antiparasitic treatments [73]. *Bixa orellana*, *Crescentia cujete,* or *Musa* spp. are also used in the treatment of demodectic mange [31]. Topical applications of Charmil gel (containing oil extracts of the plants *Cedrus deodare* and *Pongamia glabra*) have caused complete recovery in young dogs infested with *Sarcoptes scabei* var *canis* after 7 to 14 days in mild-to-moderate and severe infestations, respectively, with hair regrowth on day 28 post-treatment. No adverse reactions were observed, except for mild irritation and restlessness, which persisted for a few hours immediately after application. Compared with amitraz, Charmil gel takes longer to resolve lesions [74].

*Eclipta alba* is indicated for secondary fungal infections in dogs with mange [31].

The essential oil mixture, consisting of *Citrus limon* (1%), *Salvia sclarea* (0.5%), *Rosmarinus officinalis* (1%), and *Anthemis nobilis* (0.5%), applied locally once daily for 2 weeks, yielded excellent results in all dogs treated for *Malassezia* otitis externa [22].

Food hypersensitivity can occur at any age, and about 30% of confirmed diagnosed dogs are less than 1 year old. The combination of phytotherapeutics (*Silybum marianum*, *Glycyrrhiza glabra*, *Curcuma longa*, and *Pleurotus ostreatus*), although administered in adult animals, has been a viable therapeutic alternative in dogs with allergy-based skin manifestations [75].

### 2.4. Miscellanea

Various uses of phytotherapeutic complexes in puppies have been proposed, often anecdotally.

Dietary supplementation with *Rosmarinus officinalis* and/or *Ocimum basilicum* (0.05% powder alone, or 0.025% used in conjunction) holds promise as a nutritional management tool for the prevention and control of diabetes mellitus in dogs aged 4 to 8 months [76].

Rosemary contains volatile fractions and various phenolic compounds that not only have antihyperglycemic and anti-dyslipidemic effects, but also exert multiple biological activities, including antioxidant, anti-inflammatory, cholagogue, choleretic, and hepatoprotective properties. In infants, *Hordeum vulgare* can be used as a complementary therapy to treat neonatal jaundice [77].

Certain plants can influence dog behavior. Extracts of *Souroubea* spp. have been shown to induce physiological changes that can counter anxiety and stress [78]. Mother tincture of lemon balm (*Melissa officinalis*), valerian (*Valeriana officinalis*), and hawthorn (*Crataegus oxyacantha*), at a dosage of 1 drop/kg body weight 2–3 times daily, is often suggested for the treatment of separation anxiety, stress reduction, and the management of behavioral or psychological issues in pets. The ambient scent of lavender may be beneficial as a treatment for travel-induced arousal in dogs [79]. A recent study revealed that more than 75% of surveyed dog owners in the U.S.A. have administered cannabidiol (CBD), a phytocannabinoid found in hemp, to their pets [80], and it cannot be ruled out that some have also administered it to young dogs. The primary reason participants administered CBD to their pets was its anxiolytic and calming effects, which addressed conditions such as separation anxiety, nervousness, aggressiveness, and situations such as fireworks-related fear and travel. In contrast, it appears that most “alternative” products, including herbal remedies, are unlikely to be sufficient as a monotherapy for fear of noise, and only preventive training seems be very effective in preventing the development of fear of noise in puppies [81].

The use of herbal medicine could be considered when dealing with seizure disorders that are refractory to monotherapy in order to achieve better seizure control with minimal or no adverse effects.

*Euphrasia officinalis* may find indication in the treatment of conjunctivitis in puppies based on the findings in preterm infants with ocular discharge [82], providing relief in the presence of redness and lacrimation, improving ocular comfort.

A commercially available fish-based dog food, supplemented with glucosamine, chondroitin sulfate, fish oil-derived fatty acids, freeze-dried whole green-lipped mussel powder (*P. canaliculus*), *Boswellia serrata*, and devil’s claw (*Harpagophytum procumbens*), given to Labrador Retrievers during their first year of life did not reduce the prevalence of hip and elbow dysplasia but did result in a less severe degree of OA at 12 months of age [83].

Weaning represents a stressful period characterized by dietary and environmental changes, and puppies are more susceptible to infections, particularly *Salmonella*. Several nutraceuticals are currently employed as “alternative antibiotics” to enhance performance and gut health in animal farming. Many essential oils have exhibited antimicrobial activities that can be used in the preventive and supportive treatment of various bacterial infections in puppies, without incurring the troubling problem of antimicrobial resistance. For example, *Cymbopogon martinii* has demonstrated the highest antimicrobial activity against *Staphylococcus aureus* and *Escherichia coli*; cinnamon oil, in combination with other antimicrobial agents (such as chlorhexidine, triclosan, and gentamicin) exhibits antimicrobial activity against strains of *S. epidermidis* and *S. aureus.* Essential oils of oregano and thyme have undergone extensive testing as antimicrobial agents against numerous pathogens, including *B. cereus*, *E. coli* O157:H7, *Enterococcus faecalis*, *Listeria monocytogenes*, *S. typhimurium*, *S. aureus*, *Pseudomonas fluorescens*, and *Vibrio cholerae.* Components of essential oils, such as eugenol and isoeugenol, have demonstrated a synergistic effect with most antibiotics against *E. coli*, *E. aerogenes*, *P. vulgaris*, *P. aeruginosa*, and *S. typhimurium*, including antibiotics like ampicillin, polymyxin B, norfloxacin, tetracycline, rifampin, and vancomycin [18].

Furthermore, antioxidant treatment appears to have the potential to enhance the immune response to vaccination in young dogs [84].

### 2.5. Side Effects

Regarding herbal products, “natural” does not always equate to “safe,” especially when the patient is a puppy. The rules on the prescription, preparation, and consumption of medicinal plants are not always clear and rigid, nor are their marketing rules which differ from country to country. It is crucial to recognize that there may be risks due to the contamination of raw materials in regard to solvents used during the extraction process, as well as the presence of soil contaminants such as heavy metals, fungi, and pesticides. Additionally, accidental exposure to heat, oxygen, and light during production and storage can lead to decarboxylation, which can have serious health implications for animals consuming phytotherapeutics. This has led the American Herbal Products Association (AHPA) to propose a classification system for medicinal plants, dividing them into three safety classes [85]:

Class 1: Herbs that can be safely consumed when used appropriately (e.g., calendula, chamomile, echinacea, eyebright, hawthorn, lavender, lemon balm, nettle, peppermint, valerian, dandelion, and thistle);Class 2: Herbal plants which come with specific restrictions on use, as indicated by a qualified expert in the use of the substance. Subcategories include: *2a* (for external use only), *2b* (not recommended during pregnancy), *2c* (not advised during lactation), and *2d* (other specific use restrictions);Class 3: These herbs should only be used under the supervision of a qualified expert.

Adverse reactions to specific medicinal plants have been reported in companion animals (Table 5), so much so that the American Society for the Prevention of Cruelty to Animals (ASPCA) has reported 45 calls between 1992 and 2000 related to the accidental ingestion of products containing *Echinacea* spp. in pets [70]. Ooms et al. [86] have reported that 83% of dogs that ingested a herbal supplement containing guarana and ma huang (*Ephedra* spp.) developed signs of toxicosis (such as hyperactivity, tremors, convulsions, behavioral changes, vomiting, tachycardia, hyperthermia), and 17% of intoxicated dogs died. The estimated doses of guarana and ma huang range from 4.4 to 296.2 mg/kg body weight and 1.3 to 88.9 mg/kg body weight, respectively.

Additionally, adverse reactions, including those severe enough to cause pet death, have been associated with the use of natural flea products, particularly in kittens, in various formulations such as sprays, shampoos, and spot-ons [87].

**Table 5 vetsci-11-00133-t005:** Adverse reactions of some medicinal plants in pets [72,88,89].

Medicinal Plants (Scientific Name)	Adverse Reactions
*Aconitum* spp.	Salivation, nausea, emesis, cardiac arrhythmias
*Allium sativum*	Antiplatelet effect, hematologic disorders
*Artemisia absinthum*	Convulsions, trembling of the limbs, digestive disorders, thirst, paralysis, death
*Digitalis* spp.	Gastrointestinal upset, dizziness, weakness, muscle tremors, miosis, potentially fatal cardiac arrhythmias
*Echinacea* spp.	Hepatotoxicity
*Ephedra* spp. or Ma Huang	Hyperactivity, tremors, seizures, behavior changes, vomiting, tachycardia, hyperthermia
*Larrea tridentate*	Hepatotoxicity
*Juniperus sabina*	Gastrointestinal and respiratory disorders, haemorrhages
*Marshmallow* root	Hypoglycemic effect
*Mentha piperita*	Hepatotoxicity
*Rubus idaeus*	Reproductive disorders

Special care should also be taken when using combinations of herbal medicines with allopathic drugs for possible interactions (synergistic or antagonistic), especially in young subjects [72,88,90]. For example:-*Arnica montana* (with a mild anticoagulant effect): when combined with NSAIDs (meloxicam, phenylbutazone, or acetylsalicylic acid), it can induce potentially fatal gastric or intestinal bleeding;-Bilberry (*Vaccinium myrtillus*): caution is advised to avoid concomitant use of aspirin and other NSAIDs;-Black currant (*Ribes nigrum*): it has an additive diuretic effect with other diuretic drugs.-*Echinacea*: not recommended in combination with acetaminophen as it increases the risk of liver toxicity;-Garlic (*Allium sativum*): it should be avoided simultaneously with anticoagulants;-Ginkgo (*Ginkgo biloba* L.): it may present an additional risk of bleeding if given together with NSAIDs. It induces omeprazole hydroxylation;-Ginger (*Zingiber officinale*): it appears to have some benefit against motion sickness in dogs but reduces platelet aggregation through inhibition of thromboxane synthase. It may increase bleeding tendency if taken concurrently with aspirin or other NSAIDs;-Ginseng (*Panax gingseng*, *P. quinquefolius*): a possible interaction with imatinib has been reported;-Kava (*Piper methysticum*): in combination with acetaminophen, it can potentially increase the risk of hepatotoxicity. It also increases barbiturate-induced sleep time in laboratory animals and anticonvulsant effects in humans;-Licorice root (*Glycyrrhiza glabra*): contains plant constituents that inhibit the renal activity of 11-hydroxysteroid dehydrogenase, thereby reducing the conversion of cortisol to cortisone, resulting in increased renal levels of cortisol available to bind to mineralocorticoid receptors. It also causes a reduction in salicylate concentration. It contains high levels of potassium that can cause sodium–potassium imbalance, leading to cardiac arrhythmias and hypertension. It is contraindicated in type I diabetes;-St. John’s Wort (*Hypericum perforatum*): it should not be used concomitantly with central nervous system antidepressants. It also seems that it can reduce the clearance and increase the plasma concentrations of a number of clinical drugs including cyclosporine, midazolam, methadone, imatinib, tacrolimus, digoxin, and theophylline;-Milk thistle (*Sylibum marianum*): it inhibits the metabolism of losartan and increases the clearance of metronidazole;-Valerian (*Valeriana officinalis*): it is expected to potentiate the sedative effects of opioids.

## 3. Discussion

Therapeutic approaches in puppies are often based on extrapolations from studies conducted in adults due to the lack of pediatric-specific data or data available in children. Unfortunately, even in human pediatrics, the explanatory power extrapolated from bibliographic data is rather heterogeneous and fragmented [91], although approximately 9% of newborns are treated with herbal supplements from the first month of life, in particular for mild neonatal disorders such as flatulence, teething, or colds [92].

Puppies should not be considered as “small adults”, but they constitute a heterogeneous group, ranging from the neonatal period through the weaning stage to one year of age, and possess complex physiologic, developmental, and pharmacologic characteristics that differ from those of adults. Additionally, these characteristics can vary within dogs, not only by age group, but also by size and breed. While life stage divisions are somewhat arbitrary, they do provide a framework for creating an individualized plan to allow for preventive care specific to each dog’s needs at the appropriate time of life [93].

The development of appropriate therapeutic formulations for puppies is progressing slowly, and specific preparations for young animals are not currently available. When administering pharmaceutical preparations, it is crucial to consider the particularities of pediatric patients, especially when the parenteral route must be used. Whenever possible, the oral route is preferred in puppies. Herbal medicines are well-suited for oral administration, although not all oral preparations are equally easy to administer. Younger dogs may have difficulty swallowing tablets and crushing tablets, and opening capsules and dispersing them in liquids can compromise the palatability and bioavailability of the herbal drug, affecting its effectiveness. Currently, the nutraceutical industry is making significant investments in the veterinary field, to the extent that the use of nutraceuticals and herbs/botanicals has become more popular in the context of animal health and diseases compared to human medicine [94]. However, most marketed drugs do not specify their use in pediatric patients. Objectively, challenges persist in evaluating the pharmacokinetic characteristics of a phytocomplex, but this should not overshadow the assessment of the true efficacy and tolerability of herbal preparations, despite the ongoing controversy surrounding the applicability of evidence-based medicine and the diverse array of theories and practices commonly known as CAM [95].

## 4. Conclusions

Several studies have shown the promising use of natural ingredients in disease prevention and therapy, and herbal medicine fully falls under the so-called “pediatric integrative medicine”, filling important gaps in pediatric care. Functional foods and natural bioactive compounds have become relevant research topics in veterinary medicine, as the long-term use of traditional drugs causes complications, especially in young animals. Furthermore, the greater use of phytotherapeutic formulations in veterinary medicine would, always in the long term, favorably affect the impact of veterinary drugs on the environment, further strengthening the role of the phytotherapeutic approach within the” One Health” concept.

However, there are some problems regarding the use of natural products in veterinary pediatrics that can be summarized in the scarcity of scientific publications; in the fact that “natural” does not always correspond to “harmless”; by the risk of raw material contamination by extraction solvents; in soil contaminants such as heavy metals, fungi, and pesticides; in accidental exposure to heat, oxygen, and light during processing and storage; due to the limited knowledge about interactions, especially when combining phytocomplexes or phytocomplexes with synthetic molecules; in legislative gaps existing in some countries; and, last but not least, the lack of academic training provided by veterinary faculties.

In Italy, unlike other European countries, over 95% of medicinal plant products are marketed as food supplements, and they are therefore subject to legislative regulations in the food sector, contributing to the growing trend of use of phytotherapeutic supplements by pet owners for the following: their easy accessibility as no veterinary prescription is necessary, and therefore without any veterinary recommendation; the desire for self-medication and the now deep-rooted perception among people that herbs are effective, safer, and less expensive than conventional medicine.

An exception is made for substances containing CBD for oral intake which in Italy, from 4 September 2023, must follow the regulatory regime for narcotic drugs for prescription. However, it is worth noting that the use of pediatric phytotherapy is a valuable tool as an alternative or in combination with allopathic medicine.

A limitation of this exploratory review is the relatively narrow focus on a specific topic. One aspect that has received little attention is the efficiency and appropriateness of treatment protocols in different clinical scenarios involving young dogs. Thus, the challenge for future research is to increase methodological rigor, allowing for a comprehensive evaluation of research by encouraging new randomized controlled trials.

## 5. Future Directions

On the basis of what has been reported, it is hoped that in the near future some actions will be undertaken, aimed at improving knowledge relating to the phytotherapeutic approach in pediatric age. In particular, with regard to veterinary medical prescription, it would be advisable to fill the legislative gap existing in the European context by standardizing it between the various member states, but also to include in academic programs the study of phytotherapy, which nowadays is gaining more and more space in veterinary pharmaceutical handbooks. Furthermore, research in the pediatric field should be encouraged, both for synthetic drugs and for phytotherapeutic ones, in order to implement the currently particularly scarce scientific knowledge. Such studies should be extended at least to the most common animal species (kittens, foal, ect.) and not limited only to canines. Puppies deserve to be treated with high-quality medicines and with therapeutic protocols based on solid scientific information.

## Figures and Tables

**Figure 1 vetsci-11-00133-f001:**
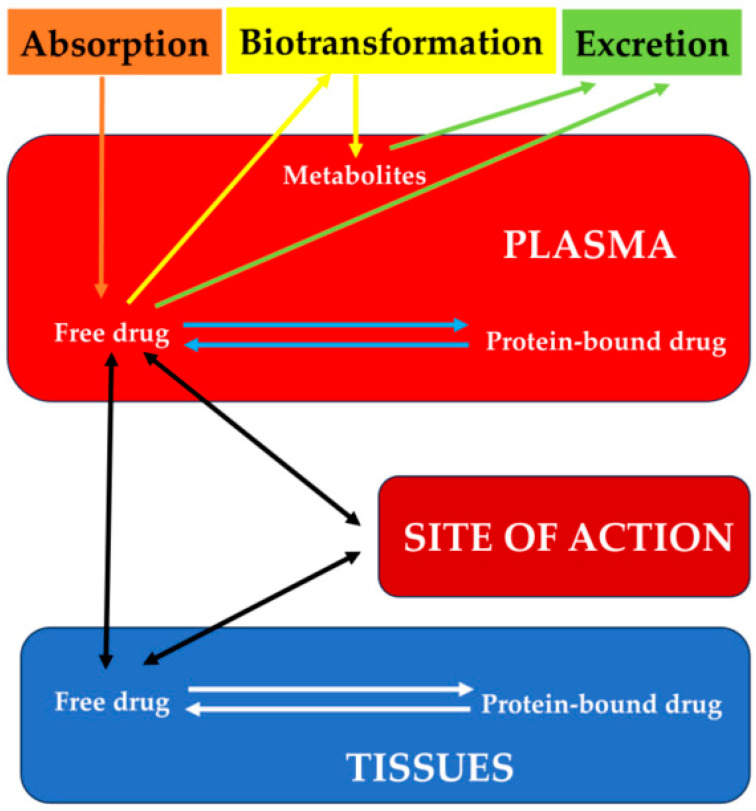
Some of the physiological factors that influence the concentration of a drug at its point of action. The figure does not represent the mechanisms of passage of drugs through biological membranes, which constitute an important pharmacokinetic factor in the absorption, distribution, and elimination of the drugs.

**Table 1 vetsci-11-00133-t001:** Examples of plant-based drugs.

PLANT	MOLECULES
*Aesculus hippocastanum*	Escin
*Ananas sativus*	Bromelain
*Artemisia annua*	Artemisinin
*Atropa belladonna*	Atropine, Hyoscine methylbromide (hemisynthesis)
*Camellia sinensis*	Theophylline
*Cannabis sativa*	Cannabidiol
*Catharanthus roseus*	Vincristine, Vinblastine, Videsine, and Vinorelbine (hemisynthesis)
*Chinchona officinalis*	Quinidine
*Claviceps purpurea*	Ergotamine, Ergotoxine, Nicergoline (hemisynthesis), Bromocriptine (hemisynthesis)
*Digitalis lanata* and *D. purpurea*	Digoxin, Digitoxin, and Methyldigoxin
*Juniperus communis*	Teniposide and etoposide
*Mappia foetida* Miers.	Comptothecin analogs
*Papaver somniferum*	Morphine, Papaverine, Codeine, and Buprenorphine
*Pausinystalia yohimbe*	Yohimbine
*Pilocarpus jaborandi*	Pilocarpine
*Podophyllum emodi* Wall.	Podophyllotoxin
*Rauwolfia serpentina*	Ajmaline and Reserpine
*Taxus brevifolia*	Paclitaxel and Taxotere (hemysinthesis)
*Vinca minor*	Vincamine

**Table 2 vetsci-11-00133-t002:** Most frequent diseases in puppies up to 12 weeks of age [42,43].

Infectious Diseases	Non-Infectious Diseases
*Bacterial infections* Local infection (e.g., skin, eyes, umbilicus) General bacterial infection Sepsis/septicemia *Viral infections* Canine adenovirus 2 Canine herpesvirus Distemper virus Kennel cough complex Canine viral enteritis (parvovirus, coronavirus, rotavirus, etc.) *Parasites* Protozoa (Giardia, Coccidia) Roundworm, hookworm	Genetic diseases Hemorrhage (vit. K deficiency) Juvenile hypoglycemia, dehydratation Hypothermia Juvenile cellulitis Impetigo Malformations, defects (e.g., swimmer puppy syndrome) Non-infectious diarrhoea Respiratory distress syndrome (RDS) Toxic milk syndrome Fading puppy syndrome Traumatic insults/injuries Foreign body ingestion or electrical cord injury Fatty liver syndrome Passive immunity transfer failure

**Table 3 vetsci-11-00133-t003:** Physiological factors contributing to disparities in oral drug absorption between pediatric and adult populations [41,49,50,51,52,53].

Gastric pH	Acid secretion from the stomach is delayed for several days after birth. In puppies, compared to adult dogs, gastric pH is less acidic.
Gastric Emptying (closely related to the physical characteristics of the ingested food)	Antral contractions in puppies increase from 0.2 contractions per minute on the day of birth to a peak of 2.3 contractions per minute on the 11th day, after which they gradually decline. Gastric emptying plays a crucial role in determining the initiation of drug absorption, as it represents a rate-limiting step preceding the exposure of drugs to the absorptive membrane of the small intestine. Gastric emptying can exhibit significant variations during growth, and studies assessing gastric emptying using radiopaque markers in dogs have not revealed any significant differences between males and females.
Bile Secretion (0.5 mL/kg/h)	It progressively develops, thereby restricting the absorption of fat-soluble substances.
Splanchnic Blood Flow	Food intake induces an increased splanchnic blood flow, which in turn will increase the absorption and transfer of nutrients into the bloodstream.
Gastrointestinal Transit Times	Puppies exhibited a shorter mean T50 compared to adults. Age did not significantly affect the mean small intestinal transit time in any breed, and the mean orocecal transit time decreased significantly only during the growth of large-breed dogs. Intestinal blood supply is lower in puppies. Food intake can influence the disintegration of formulations and drug dissolution in the GI tract. The impact of food on drug absorption depends on the dosage form’s nature, the excipients utilized in the formulation, and the particle size of the drug in the formulation. This is especially relevant in younger patients, where feeding occurs more frequently than in adults. Conspicuous jejunal lymph nodes and a mild amount of anechoic peritoneal fluid were considered normal.
Membrane Interactions	Immediately after birth, the special epithelium starts to disappear essentially gone after 24 h. Nevertheless, during the first 2 days of life, systemic effects may occur following oral administration of drugs that are not normally absorbed from the intestine. Drug absorption in pediatric patients involves transporters and enzymes that may not be fully mature. High viscosity within the intestinal lumen can slow down the diffusion rate of a drug, leading to reduced overall absorption.
Intestinal Microbial Flora	The gut microbiome of dogs is more like that of humans than that of mice and pigs. During weaning, puppies’ gut microbiota gradually becomes more similar to that of adult dogs due to the transition from milk to solid food, influenced by both dietary and behavioral factors. This microbiome development can have consequences on enteric metabolism and the intestinal wall. The predominant phyla in feces of puppies, pregnant, and lactating bitches are *Firmicutes*, *Bacteroidetes*, *Fusobacteria*, and *Actinobacteria*. Various factors, including breed, age, living conditions, diet, and methodology, can contribute to this variability. Older age was associated with a lower proportion of *Fusobacteria*.

**Table 4 vetsci-11-00133-t004:** Major physiological factors contributing to disparities in drug metabolism and elimination between pediatric and adult populations [40,43,47,49,52,53,54].

Liver	Immaturity of the liver results in reduced drug clearance in puppies. The hepatic microsomal pathways associated with drug metabolism develop rapidly during the first 3 to 4 weeks after birth, and by 8 to 12 weeks, they approach activity levels similar to those in adult animals. The in vitro activities of enzymes like P-450, glucose-6-phosphatase (G6P), and UDP-glucuronyl transferase (GT) are immature at birth and develop gradually during postnatal life. Because phase I (oxidation) and phase II (glucuronidation) hepatic enzyme systems are not fully functional in puppies, drugs that require hepatic metabolism for excretion tend to reach higher plasma levels. Conversely, drugs that rely on hepatic metabolism for activation have lower plasma concentrations. Additionally, oral drugs subject to first-pass metabolism are at risk of accumulating to toxic levels in plasma if administered at the adult dose. The immaturity of the liver can exacerbate many coagulopathies during the pre-pubescent period.
Kidney (renal blood flow: 440 mL/min/m^2^)	Renal function in puppies appears to mature within the first 4 to 6 weeks of life. The volume of nephron segments continues to grow from postnatal week 2, when nephrogenesis ceases, to approximately postnatal week 28, resulting in an enlargement of up to 300%. This maturation process leads to a reduced renal clearance of water-soluble drugs, primarily due to the low glomerular filtration rate and renal blood flow in neonates, and later, a reduced renal excretion caused by immature renal tubules. Incomplete tubular absorption is responsible for glucosuria in puppies younger than 8 weeks of age. During the first 8 weeks, urine-specific gravity varies from 1.006 to 1.017.

## Data Availability

No new data were created or analyzed in this study. Data sharing is not applicable to this article.

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
