# Peer review of "Phytotherapeutic Approaches in Canine Pediatrics"

_vetsci, 2024, doi:10.3390/vetsci11030133_

Round 1
Reviewer 1 Report
Comments and Suggestions for Authors
This review is a literature review on the Phytotherapy, the use of botanical remedies. I is an ancient branch of medicine that is making a comeback. However, there is a lack of research on its use in pediatric canine medicine. Dog owners often turn to these remedies due to their accessibility and perception of safety, often bypassing professional advice. This review aims to highlight the current literature on phytotherapy in canine pediatrics, particularly focusing on the adverse effects and the need for more research in this area.
A first review of the use of alkaloids contained in plants is made, then a description of the limitations of puppies to be treated with medicines compared to adults and finally a description of the use of different plants to treat digestive, respiratory, dermatological and other pathologies.
The adverse effects of plants, as natural products but not always to be considered safe, are explained with reference to the classification proposed by the American Herbal Products Association.
It concludes that the use of natural ingredients in disease prevention and therapy is promising, particularly in paediatric integrative medicine and veterinary medicine. However, the use of natural products in veterinary paediatrics poses some challenges, such as the scarcity of scientific publications, possible contamination of raw materials and limited knowledge about interactions. The use of paediatric herbal medicine is considered a valuable tool in combination with conventional medicine. Future research should focus on increasing methodological rigour and conducting more randomised controlled trials to ensure high quality treatment protocols for young animals.
The main strength of this review is that it provides an ancestral view of the use of plants to cure diseases, as it was done centuries ago, but uses today's concepts of evidence-based medicine.
The main weakness is that the studies on which this evidence is based come from treatments in adult patients and it is assumed that the physiology of the neonate or young animal must be a barrier to their use, with a lack of scientific evidence in this respect. However, the author states throughout the review that a better understanding of the effects of these herbs in young animals is needed.
Specific comments.
please review the text of the tables which have a size and typology that should be reduced and adapted.
Conclusion
This article is suitable for publication.
Author Response
Thanks for your comments and suggestions.
I divided table 3 into two parts (one relating to absorption and the other to elimination) and I also inserted a figure also intervening on the text (highlighted in yellow).

Reviewer 2 Report
Comments and Suggestions for Authors
I really like the work. It is impresive how many hebal medicine can be used and not used with puppies i would suggest in the conclusion deeper investigation in this field not only in puppies, kitten and adults individuos.
maybe you should give more importance to the fact that can be bought wuthout any veterinarian recommendation and maybe for veterinarian ther is a lack of knowledge in this field.
I only miss references in table 3. you give a lot of information about physiological factor but do not reference any of them
in line 96 you make a comment about the femail veterinarians working in clinics. this comment can be sexist if you do not justified it. i suggest to avoid this kind of comment.
Author Response
Thank you for your welcome comments and valuable suggestions.
Table 3 has been resized with the creation of another table to make it (at least I hope) more editorially usable. For both I provided the bibliographical references that allowed their creation.
in line 96 you make a comment about the femail veterinarians working in clinics. this comment can be sexist if you do not justified it. i suggest to avoid this kind of comment. The bibliographic reference is reported in the text. Among other things, this is a data recently published in a Q1 magazine (Romero et al. Front. Vet. Sci. 2022, 9, 1060738. doi: 10.3389/fvets.2022.1060738). I have reformulated the sentence (highlighted in yellow). However, if you don't think it's appropriate because it's "not politically correct", I have no difficulty in eliminating it.
